# DIRECT PREFERENCE OPTIMIZATION FOR DYNAMICAL SYSTEM MODELING

## ABSTRACT

Deep learning models for dynamical system forecasting, despite their success, often falter when trained solely on pixel-wise numerical metrics. This paradigm leads to overly smooth predictions that fail to capture high-impact, rare events and lack the physical plausibility demanded by domain experts. To bridge this gap, we introduce `PRISM`, a novel human-machine collaborative framework that aligns predictive models with human preferences for physical realism and perceptual quality. `PRISM`'s core mechanism involves distilling complex, often non-differentiable human judgments into a differentiable preference model. This is achieved by training on prediction pairs generated via a diverse sampling strategy and ranked by human-trusted proxy metrics. Subsequently, this learned preference oracle is used to fine-tune the foundational predictive model through a joint optimization process, which we theoretically ground as a bi-level optimization problem converging to a stable equilibrium. Extensive experiments on challenging benchmarks in fluid dynamics and numerical weather forecasting demonstrate that `PRISM` serves as a versatile, plug-and-play enhancer for a wide range of state-of-the-art models. It consistently yields predictions that are not only numerically accurate but also qualitatively superior in capturing critical physical phenomena and visual coherence. Codes are available at https://anonymous.4open.science/status/PRISM_main-CC1D.

## 1 INTRODUCTION

In contemporary scientific research and engineering applications, modeling and predicting complex dynamic systems serve as important tools to understand and reveal physical phenomena. They are widely applied in areas such as weather forecasting, climate change prediction, and fluid dynamics (Wu et al.; 2024d; 2023; Bi et al., 2022). Precise dynamic prediction helps us better comprehend natural laws and provides scientific support for disaster prevention, resource management, and major engineering decisions. However, most dynamic system modeling methods primarily optimize numerical metrics (Li et al., 2020; Wu et al., 2024c). While they strive to minimize overall risks, they often fail to accurately capture rare events like extreme weather and sudden fluid vortices. Worse still, these methods tend to neglect the perceptual consistency and are not informative in expressing the physical interpretability of predicted results (including visualizations).

To conquer the aforementioned shortcomings, researchers propose various improvement strategies. First, models based on multi-scale feature extraction make significant progress in learning the integration of local and global spatiotemporal information (Wu et al., 2024e; 2023; 2024f; He et al.). By hierarchically extracting key features at different scales, they enhance the ability to capture local extreme phenomena to some extent. However, these methods often lead to more complex model structures and higher computational costs. Second, solutions based on generative adversarial networks (GAN) (Goodfellow et al., 2014) or energy models attempt to generate diverse prediction scenarios by approximating the real distribution, addressing higher-order uncertainties that average loss cannot cover (Zhang et al., 2023; Ravuri et al., 2021; Wang et al., 2023). However, the adversarial training process itself is unstable, prone to mode collapse or gradient oscillations (Li et al., 2018; Thanh-Tung & Tran, 2020), resulting in insufficient reliability in predicting extreme or rare scenarios.

In this context, some studies integrate *physical constraints* and *human prior knowledge* into predictive models. For example, in numerical weather forecasting, some work embeds physical laws directly

Figure 1: **An overview of Benchmark**. (1) First, obtain the raw data. (2) Then, pretrain advanced operator learning methods or spatiotemporal forecasting models. (3) Next, provide different prompts for the pretrained model in the second stage. Here, the prompt is not text but Perlin noise, commonly used in scientific computing. By applying noise with varying intensities, the model generates diverse results, which are then scored using high-quality, non-differentiable human preference metrics. The illustration uses a turbulence energy spectrum, and the final preference dataset is constructed by selecting the highest- and lowest-scoring results.

into the network structure or introduces physical corrections in the post-processing stage to ensure that the output results comply with conservation laws of energy and momentum (Zhang et al., 2023; Rao et al., 2023; De Bézenac et al., 2019; Raissi et al., 2019; Jagtap et al., 2020). However, numerical optimization and physical constraints cannot easily achieve a unified goal through simple weighted sums. The scales of physical constraints and applicable scenarios vary greatly, and boundary conditions are extremely complex (Fadlun et al., 2000; John & Anderson, 1995; Efendiev & Hou, 2009). This makes it difficult for models to maintain the same stability and adaptability across different fields and environments. Additionally, using only numerical metrics, such as MSE, to measure prediction quality ignores human needs for interpretability, visual perceptual consistency, and attention to extreme events. Especially in highly sudden dynamic processes, like extreme weather (Racah et al., 2017; Wu et al., 2024f) and fluid turbulence (Wang et al., 2020; Liu et al., 2020), experts focus more on accurately characterizing the overall structure, evolution trends, and underlying mechanisms rather than minimizing point-to-point errors.

*Therefore, turning human preferences or approvals for prediction results into learnable metrics and performing end-to-end model optimization remains a key challenge.*

To address this issue, we propose a unified modeling framework `PRISM` that combines numerical accuracy with human preference scores, based on human preference learning (Rafailov et al., 2024) and diverse sampling (Bhattacharyya et al., 2018; Ma et al., 2021). Specifically, we first use risk error (like MSE) during the pre-training phase to ensure overall numerical consistency between the predictions and the ground-truth. Then, by adding perturbations (Chen et al., 2024; Hu et al., 2023) to the input or replacing discrete embeddings (Van Den Oord et al., 2017), we generate a diverse set of prediction samples and select high- and lower-quality prediction pairs based on human-trusted metrics, such as physical consistency (Wu et al., 2024f; Wang et al., 2020), visual structure similarity (Hore & Ziou, 2010), or domain-specific preference evaluations. Going beyond this, we train a preference model that learns to rank different predictions under the same input conditions. Finally, we jointly optimize the preference model with the base prediction model: while maintaining numerical prediction accuracy, we explicitly update the models in directions that better align with human *preferences* or *interpretability*.

To validate our concept, we construct an open-source scientific dataset that integrates human preferences **within the first shot**. To solve this, we collaborate with physics experts and use crowdsourced annotations to build a dataset called *HPSci* (Human Preference for Scientific Computing). *HPSci* covers typical dynamical system scenarios such as turbulence, Rayleigh-Bénard convection and fire spread, providing rich prediction samples annotated with human preferences. Build on this, we can explore deeply how to combine human preferences with physical consistency to enhance the performance and interpretability of dynamical system prediction. In Section 2, we will introduce the construction method and characteristics of this dataset.

In summary, the contribution of our paper can be summarized as follows: (1) *Novel Methodology.* We construct a multi-objective optimization framework that combines numerical loss with human preference scores. This provides a flexible and adjustable unified training scheme for various application scenarios. (2) *New Strategy.* In sampling strategies and preference Benchmark construction, we generate a diverse set of candidate predictions by perturbing inputs or replacing discrete embeddings. We then select positive and negative sample pairs based on human-trusted metrics, effectively enhancing the ability to perceive extreme or abnormal scenarios. (3) *Superior Performance.* We analyze the convergence and optimality of our method using a bi-level optimization and game theory approach. We validate its superior performance in real weather forecasting and fluid simulation tasks.

## 2 BENCHMARK

Existing literature does not have a public dataset that combines scientific computing features with human preferences. To solve this, we work with physics researchers and use crowd-sourced annotations to create *HPSci*, a *H*uman *P*reference dataset for *Sci*entific computing. The main process is shown in Figure 1.

First, we choose typical dynamical system scenarios, such as fluid turbulence, Rayleigh-Bénard convection, and wildfire spread, from scientific computing datasets like BLAST-Net (Chung et al., 2024) and PDEBench (Takamoto

Table 1: Comparison among different Benchmarks.

| Method | Multi-Objective | human preference | 3D |
|---|---|---|---|
| PDEBench (Takamoto et al., 2022) | ✗ | ✗ | ✗ |
| BLAST (Chung et al., 2024) | ✗ | ✗ | ✓ |
| EAGLE (Janny et al., 2023) | ✗ | ✗ | ✗ |
| Prometheus (Wu et al., 2024b) | ✗ | ✗ | ✗ |
| **HPSci (Ours)** | ✓ | ✓ | ✓ |

et al., 2022). Then, we use pretrained forecasting models, such as FNO and ConvLSTM, to make various forecasts. Next, to increase uncertainty and variety in the model forecasts, we add different changes, including Gaussian noise and discrete embedding replacements, to the input or intermediate features to create more candidate forecasts. We then use physical consistency measures, like turbulent kinetic energy and energy conservation, to filter these samples. To gather human preferences, we use both crowdsourcing and expert annotations. Given the same input scenario and multiple forecast outputs, annotators choose or rate the forecasts based on their perceived quality, creating positive and negative pairs for preference learning. Finally, we organize these annotated samples with the original inputs and observed results to create *HPSci*, which we release publicly. This dataset provides a base for future research on combining human preferences and physical consistency in dynamical system forecasting.

Table 1 summarizes the benchmark details. Compared to traditional benchmarks, the HPSci benchmark offers three key advantages: *multi-objective optimization*, *human preference integration*, and *comprehensive 3D modeling support*. It combines numerical accuracy with human-perceived quality and supports complex 3D dynamical system forecasting, improving model practicality and reliability.

## 3 METHODOLOGY

We propose `PRISM`, a novel framework for dynamical system modeling. Its core objective is to transcend the limitations of conventional pixel-wise numerical metrics, such as Mean Squared Error (MSE), by aligning model predictions with human-perceptible quality and physical intuition. As illustrated in Figure 2, the `PRISM` architecture follows a meticulously designed three-stage optimization pipeline: (1) ***Fidelity-Focused Pre-training of a Foundational Model***, which establishes a robust baseline for dynamical forecasting; (2) ***Distillation of a Human Preference Oracle***, which translates complex, often non-differentiable expert criteria into a smooth, differentiable scoring function; and (3) ***Policy Fine-tuning via Direct Preference Optimization***, which leverages the learned preference information to guide the refinement of the foundational model, thereby achieving a Pareto-optimal balance between numerical accuracy and perceptual realism.

### 3.1 PROBLEM FORMULATION

We first formalize the task of spatiotemporal forecasting. Given a sequence of historical state observations of a dynamical system, represented as a tensor $X \in \mathbb{R}^{T_{in} \times C \times H \times W}$, our goal is to

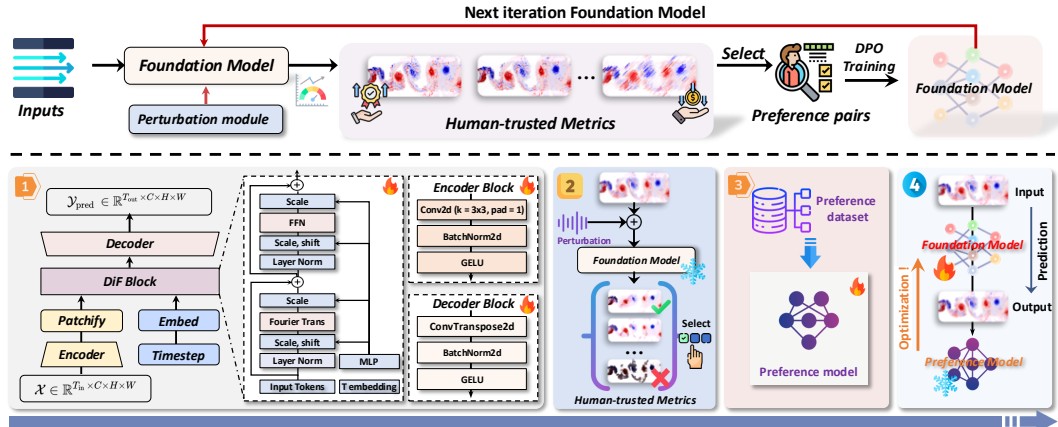

Figure 2: **An overview of** `PRISM`. Our method consists of four steps: *(1)* Pretrain the base model and optimize it using mean squared error (MSE) loss. *(2)* Generate diverse prediction samples with a perturbation module and evaluate their quality using human-trusted metrics. *(3)* Create a preference dataset and train a preference model to assess prediction quality. *(4)* Combine preference model scores with base model accuracy to jointly optimize the prediction model, achieving both accuracy and alignment with human preferences.

predict its future state sequence $Y_{\text{true}} \in \mathbb{R}^{T_{out} \times C \times H \times W}$. Here, $T_{in}$ and $T_{out}$ denote the temporal lengths of the input and output, $C$ is the number of channels representing physical fields, and $H \times W$ is the spatial grid resolution.

We define our predictive model as a deep neural network $f_\theta : X \mapsto \hat{Y}$, parameterized by $\theta$. In the standard supervised learning paradigm, the optimization objective is to minimize the expected loss over the data distribution D, which is typically the Mean Squared Error (MSE):

$$L_{\text{MSE}}(\theta) = \mathbb{E}_{(X, Y_{\text{true}}) \sim D} \left[ \| f_\theta(X) - Y_{\text{true}} \|_F^2 \right], \tag{1}$$

where $\| \cdot \|_F$ denotes the Frobenius norm. However, this loss function tends to penalize errors across all frequencies uniformly, causing the model to learn a conditional mean. The resulting predictions are often overly smooth and fail to capture high-frequency details, extreme events, or critical topological structures.

Our central hypothesis is that an ideal prediction $\hat{Y}$ should maximize an implicit utility function $U(\hat{Y}, Y_{\text{true}})$, which encapsulates not only numerical similarity but also human preferences regarding high-level semantics such as physical consistency and structural plausibility. As $U$ is latent and non-differentiable, our task is to construct a tractable surrogate objective for its optimization.

### 3.2 STAGE 1: FOUNDATIONAL MODEL PRE-TRAINING FOR NUMERICAL FIDELITY

To ensure the model first masters the fundamental evolutionary laws of the system, we conduct an initial pre-training stage for the predictive model $f_\theta$. The objective here is to minimize the $L_{\text{MSE}}$ defined in Equation 1, thereby obtaining a set of initial parameters $\theta_0$ that exhibit strong numerical stability and foundational predictive capabilities:

$$\theta_0 = \arg\min_\theta L_{\text{MSE}}(\theta). \tag{2}$$

As shown in Figure 2, our foundational model $f_\theta$ employs a hybrid **Encoder-Diffusion-Decoder** architecture. The input X is first mapped into a latent space by an encoder $E_{\theta_{\text{enc}}}$. Subsequently, a Transformer-based diffusion model (DiT), $D_{\theta_{\text{dit}}}$, denoises and refines the latent representations to capture global spatiotemporal dependencies. Finally, a decoder $G_{\theta_{\text{dec}}}$ reconstructs the processed latent representation into a high-resolution prediction $\hat{Y}$. The entire forward pass is a function composition:

$$\hat{Y} = f_\theta(X) = G_{\theta_{\text{dec}}}(D_{\theta_{\text{dit}}}(E_{\theta_{\text{enc}}}(X))). \tag{3}$$

This pre-training stage provides a high-quality "reference policy" for the subsequent preference alignment.

### 3.3 STAGE 2: DISTILLATION OF A HUMAN PREFERENCE ORACLE

The core task of this stage is to translate abstract human preferences into an operational, differentiable signal. We achieve this by training a separate **Preference Model** $S_\phi$, parameterized by $\phi$, designed to output a scalar score for any given prediction $\hat{Y}$. A higher score indicates greater alignment with human preferences.

#### 3.3.1 EXPLORATORY CANDIDATE GENERATION

To train a discerning preference model, we require a diverse set of candidate predictions for the same input X, exhibiting variations in quality. We generate these candidates by introducing random perturbations at the input of the pre-trained model $f_{\theta_0}$, thereby exploring the neighborhood of the conditional probability distribution $p(Y|X)$. Specifically, we sample a perturbation $\delta_i$ from a predefined distribution $P_\sigma$ (e.g., a Gaussian distribution $\mathcal{N}(0, \sigma^2 I)$) to generate a set of $N$ candidates, $\mathbb{Y}_X$:

$$\mathbb{Y}_X = \{\hat{Y}_i\}_{i=1}^N \quad \text{s.t.} \quad \hat{Y}_i = f_{\theta_0}(X + \delta_i), \quad \delta_i \sim P_\sigma. \tag{4}$$

This step is crucial, as it provides a rich and varied dataset for preference learning, covering a spectrum of outcomes from structurally coherent to artifact-laden.

#### 3.3.2 MAXIMUM LIKELIHOOD ESTIMATION OF THE PREFERENCE MODEL

Next, we construct a preference dataset $D_{\text{pref}}$. As depicted in the "Select" step of Figure 2, we employ a set of domain-specific, widely accepted **human-trusted metrics** $M(\cdot, \cdot)$ as an automated proxy for expert judgment. For instance, in fluid dynamics, M could be the similarity of the turbulent kinetic energy spectrum; for weather forecasting, it could be the Critical Success Index (CSI).

For each candidate set $\mathbb{Y}_X$, we select a "winner" $Y_w$ and a "loser" $Y_l$ to form a preference pair $(Y_w, Y_l)$, where $M(Y_w, Y_{\text{true}}) > M(Y_l, Y_{\text{true}})$. We assume that human preferences follow the **Bradley-Terry model**, wherein the probability of preferring $Y_w$ over $Y_l$ is proportional to the exponent of their latent rewards:

$$p(Y_w \succ Y_l) = \frac{\exp(r^*(Y_w))}{\exp(r^*(Y_w)) + \exp(r^*(Y_l))} = \sigma(r^*(Y_w) - r^*(Y_l)), \tag{5}$$

where $r^*(\cdot)$ is the true, unknown reward function, and $\sigma(\cdot)$ is the logistic sigmoid function. Our preference model $S_\phi$ aims to learn this reward function. We train $S_\phi$ by maximizing the log-likelihood on $D_{\text{pref}}$, with the following loss function:

$$L_{\text{pref}}(\phi) = -\mathbb{E}_{(Y_w, Y_l) \sim D_{\text{pref}}} \left[ \log \sigma \left( S_\phi(Y_w) - S_\phi(Y_l) \right) \right]. \tag{6}$$

Upon convergence, the model $S_\phi$ becomes a **differentiable preference oracle**, capable of scoring any prediction and effectively emulating the complex evaluation process of human experts.

### 3.4 STAGE 3: POLICY FINE-TUNING VIA DIRECT PREFERENCE OPTIMIZATION

In the final stage, shown as "DPO Training" in Figure 2, we fix the preference oracle $S_\phi$ and use it to guide the fine-tuning of the foundational model $f_\theta$. We treat $f_\theta$ as a **policy** $\pi_\theta$ that generates predictions, while $S_\phi$ provides the reward signal. Diverging from traditional reinforcement learning methods, we adopt the principles of **Direct Preference Optimization (DPO)**, which reframes the reward maximization problem as a simple classification task, thereby circumventing explicit reward modeling and its associated sampling instabilities.

We formulate a composite loss function to update $\theta$, which integrates numerical fidelity with preference alignment:

$$L_{\text{Total}}(\theta) = L_{\text{MSE}}(\theta) + \lambda L_{\text{DPO}}(\theta; \theta_0), \tag{7}$$

where $\lambda$ is a hyperparameter that balances the two objectives. The $L_{\text{DPO}}$ term, inspired by the core idea of DPO, directly optimizes the policy using the preference model. Its form is analogous to $L_{\text{pref}}$, but the optimization is performed over the generator's parameters $\theta$:

$$L_{\text{DPO}}(\theta; \theta_0) = -\mathbb{E}_{X \sim D, (\delta_w, \delta_l) \sim P_\sigma} \left[ \log \sigma \left( S_\phi(f_\theta(X + \delta_w)) - S_\phi(f_\theta(X + \delta_l)) \right) \right]. \tag{8}$$

Here, the loss is constructed by scoring the outputs of the current model $f_\theta$. This is mathematically equivalent to implicitly maximizing the reward of the policy $\pi_\theta$ relative to the reference policy $\pi_{\theta_0}$. The gradient update rule for the entire optimization process is:

$$\theta \leftarrow \theta - \eta \nabla_\theta \left( L_{\text{MSE}}(\theta) + \lambda L_{\text{DPO}}(\theta; \theta_0) \right). \tag{9}$$

By minimizing $L_{\text{Total}}$, we drive the predictive model $f_\theta$ to explore regions of the solution space that yield higher preference scores—and thus are more physically intuitive and visually realistic—while remaining anchored to the ground truth by the numerical fidelity term. Ultimately, the `PRISM` framework yields a dynamical system model that is both "accurate" and "perceptually superior."

## 4 EXPERIMENTS

**Benchmarks.** As presented in Table 2, we conduct experiments on three datasets. First, in the *Prometheus* dataset, following the design of (Wu et al., 2024b), we select two scenarios: tunnel fire and pool fire. Considering the significant impact of fire spread on human safety, we focus on the visual perception ability of the prediction results in this dataset and choose the Structural Similarity Index (SSIM) (Brunet et al.,

Table 2: Summary of experiment benchmarks, including the number of training samples ($N_{\text{train}}$), the number of testing samples ($N_{\text{test}}$), data dimensions (number of channels $C$, height $H$, width $W$), and the input/output time steps ($I/O$).

| DESCRIPTIONS | $N_{train}$ | $N_{test}$ | $(C, H, W)$ | $I/O$ |
|---|---|---|---|---|
| PROMETHEUS-T | 30000 | 2000 | (2, 32, 480) | 10/10 |
| PROMETHEUS-P | 30000 | 2000 | (2, 32, 64) | 10/10 |
| RAYLEIGH–BÉNARD | 1544 | 193 | (2, 64, 448) | 1/99 |
| SEVIR | 35,718 | 4465 | (1, 192, 192) | 13/12 |

2011) for optimization. Second, for *Rayleigh–Bénard convection*, we follow the design of (Wang et al., 2020) and select turbulent kinetic energy as the evaluation metric. In fluid dynamics, turbulent kinetic energy represents the average kinetic energy per unit mass associated with vortices in turbulence. From a physical perspective, we characterize it by measuring the root mean square of velocity fluctuations. Finally, to reflect extreme precipitation events, we choose the *SEVIR* (Veillette et al., 2020) dataset and, following the design of (Gao et al., 2022b), take the Critical Success Index (CSI) as the optimization metric (Schaefer, 1990).

**Backbones.** We comprehensively use about 11 backbones, including basic vision backbone: ResNet (He et al., 2016), U-Net (Ronneberger et al., 2015), Vision Transformer (ViT) (Dosovitskiy, 2020), Swin Transformer (Swin) (Liu et al., 2021), etc. Spatio-temporal Prediction model: SimVP (Gao et al., 2022a), PastNet (Wu et al., 2024e). Typical neural operators: FNO (Li et al., 2020), CNO (Raonic et al., 2024), UNO (Rahman et al., 2022) and Based on Graph Neural Network: MGN (Pfaff et al., 2020), EGNN (Satorras et al., 2021).

**Implementations.** For fairness, we set the hidden layer dimension of all models to 256 and the learning rate to 1e-3, using a cosine annealing strategy for adjustment. We choose the Adam optimizer (Kingma & Ba, 2014) and use MSE as the loss function. We conduct all experiments on servers equipped with eight 40GB NVIDIA A100 GPUs and perform inference on a single 40GB NVIDIA A100 GPU. See Appendix for comprehensive implementations.

### 4.1 MAIN RESULTS

We compare `PRISM` with multiple existing models and validate its effectiveness in improving numerical accuracy and human perceptual quality.

**Qualitative Analysis.** Table 3 shows model performance improvements with `PRISM`. On Prometheus (MSE/SSIM focus), basic CNNs gain substantially: ResNet (SSIM 0.8334→0.8485) and U-Net (0.8298→0.8643) show enhanced structural accuracy. Advanced models (SimVP/PastNet) maintain MSE while increasing SSIM, confirming better visual consistency. For Rayleigh–Bénard (MSE/MAPE focus), `PRISM` lowers MAPE: ResNet (25.66%→24.99%) and U-Net (13.72%→12.73%), improving dynamical system sensitivity. On SEVIR (MSE/CSI focus), `PRISM` boosts CSI: ViT (0.3847→0.3984) and Swin (0.3983→0.4212), demonstrating enhanced extreme weather prediction. `PRISM` achieves top performance: Prometheus (MSE 0.0287→0.0281; SSIM 0.9103→0.9233), Rayleigh–Bénard (MSE 0.1023→0.0983; MAPE 12.31%→10.29%), and

Table 3: **Performance comparison of various models with and without the `PRISM` framework.** The table presents the performance of different models in their original versions (Ori) and after applying `PRISM` (+ `PRISM`). '∗' indicates a memory overflow. '⁻' indicates that FNO failed to learn high-frequency components, yielding non-informative predictions.

| MODEL CATEGORY | PROMETHEUS | | | | RAYLEIGH-BÉNARD | | | | SEVIR (MSE IS $10^{-3}$) | | | |
| --- | --- | --- | --- | --- | --- | --- | --- | --- | --- | --- | --- | --- |
| | ORI | | + PRISM | | ORI | | + PRISM | | ORI | | + PRISM | |
| | MSE | SSIM | MSE | SSIM | MSE | MAPE(%) | MSE | MAPE(%) | MSE | CSI | MSE | CSI |
| **COMPUTER VISION BACKBONES** | | | | | | | | | | | | |
| 🖸 RESNET _CVPR'16 | 0.0982 | 0.8334 | 0.0972 | **0.8485** | 0.6990 | 25.66 | 0.6872 | **24.98** | 5.0478 | 0.3234 | 4.9873 | **0.3398** |
| 🖸 U-NET _MICCAI'15 | 0.1067 | 0.8298 | 0.0921 | **0.8643** | 0.1246 | 13.71 | 0.1372 | **12.73** | 4.1119 | 0.3593 | 4.0932 | **0.3674** |
| 🖸 VIT _ICLR'21 | 0.0713 | 0.8512 | 0.0722 | **0.8627** | 0.1354 | 14.38 | 0.1293 | **13.98** | 3.9843 | 0.3847 | 3.8475 | **0.3984** |
| 🖸 SWIN _CVPR'21 | 0.0729 | 0.8776 | 0.0708 | **0.8921** | 0.1273 | 15.74 | 0.1283 | **14.87** | 3.7463 | 0.3983 | 3.4354 | **0.4212** |
| **SPATIOTEMPORAL MODELS** | | | | | | | | | | | | |
| ⊞ SIMVP _CVPR'22 | 0.0342 | 0.9233 | 0.0326 | **0.9301** | 0.0926 | 20.00 | **0.0872** | 10.28 | 3.4632 | 0.4538 | 3.4532 | **0.4676** |
| ⊞ PASTNET _MM'24 | 0.0299 | 0.9398 | 0.0298 | **0.9421** | 0.1126 | 11.83 | 0.1112 | **11.09** | 3.3874 | 0.4695 | 3.3982 | **0.4701** |
| **OPERATOR LEARNING MODELS** | | | | | | | | | | | | |
| ♣ FNO _ICLR'21 | 0.0506 | 0.6537 | 0.0507 | **0.6538** | 0.1455 | 11.65 | 0.1203 | 10.29 | — | — | — | — |
| ♣ CNO _NEURIPS'23 | 0.0862 | 0.8654 | 0.0842 | **0.8722** | 0.1134 | 11.57 | 0.0927 | 10.28 | 4.3784 | 0.3384 | 4.3698 | **0.3409** |
| ♣ UNO _KDD'22 | 0.0499 | 0.8937 | 0.0453 | **0.9123** | 0.1173 | 11.60 | 0.1283 | 10.29 | 3.6372 | 0.4003 | 3.5947 | **0.4092** |
| **GRAPH NEURAL NETWORKS** | | | | | | | | | | | | |
| ♣ MGN _ICLR'21 | 0.1079 | 0.8421 | 0.0921 | **0.8521** | 0.2731 | 15.42 | 0.2563 | **14.28** | ∗ | ∗ | ∗ | ∗ |
| ♣ EGNN _ICML'21 | 0.1722 | 0.7829 | 0.1574 | **0.8021** | 0.7832 | 22.83 | 0.7726 | **20.09** | ∗ | ∗ | ∗ | ∗ |
| 🏆 **OURS** | | | **0.0287** | 0.9103 | **0.0281** | 0.9233 | **0.0926** | 12.30 | **0.0983** | 10.28 | **3.3623** | 0.4783 | **3.3546** | 0.4893 |

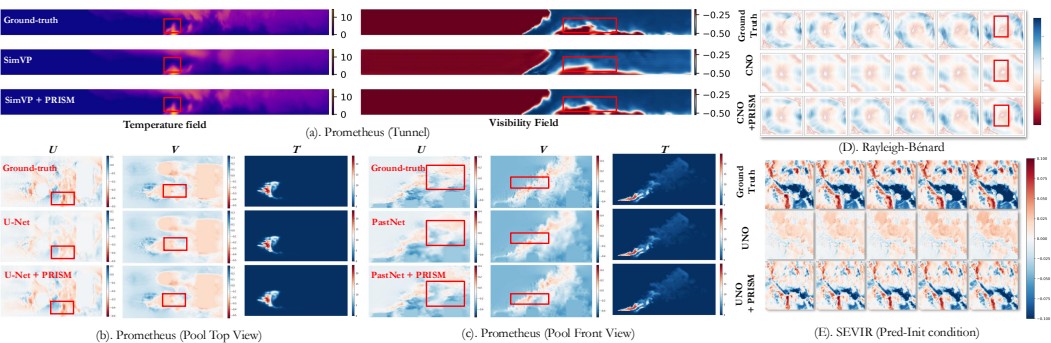

Figure 3: **Visual Results.** (a) Prometheus tunnel benchmark. Shows SimVP's temperature and visibility fields with and without `PRISM` at the final time step. (b) Prometheus pool fire benchmark (top view). Shows SimVP's U V velocity and temperature fields with and without `PRISM` at the final time step. (c) Prometheus pool fire benchmark (front view). Shows PastNet's U V velocity and temperature fields with and without `PRISM` at the final time step. (d) Rayleigh-Bénard benchmark. Shows CNO's U V velocity vectors with and without `PRISM`. (e) SEVIR benchmark. Shows UNO's U V velocity vectors with and without `PRISM`.

SEVIR (MSE 3.3623→3.3546; CSI 0.4783→0.4893). Neural operators (FNO/CNO/UNO) improve with `PRISM`, though FNO fails on high-frequency components (marked "−"). Graph networks (MGN/EGNN) show partial gains despite memory overflow issues (marked "∗").

**Quantitative Analysis.** Figure 3 visualizes `PRISM`'s enhancements: (a) SimVP with `PRISM` better captures flame paths and smoke diffusion in tunnel fires. (b-c) SimVP/PastNet with `PRISM` accurately reproduce fluid dynamics and temperature patterns in pool fires. (d) CNO with `PRISM` reveals clearer convective roll structures in Rayleigh–Bénard flows. (e) UNO with `PRISM` precisely locates heavy precipitation regions in SEVIR extreme weather prediction. Overall, Figure 3 effectively demonstrates how `PRISM` enhances the performance of various simulation models, leading to more accurate and detailed representations of complex physical phenomena.

### 4.2 MODEL ANALYSIS

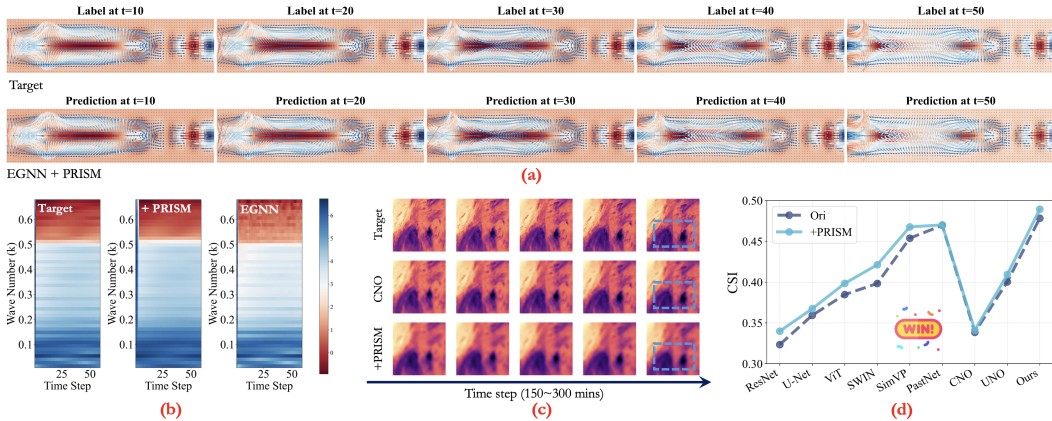

Figure 4: (a) SWE dataset forecasts, showing the performance of EGNN + `PRISM` at different time steps. (b) Energy spectrum analysis, comparing the target data, `PRISM`-enhanced approach, and EGNN model across high and low frequencies. (c) SEVIR dataset forecasts, demonstrating the improvements of the `PRISM`-enhanced approach in thunderstorm forecasting. (d) CSI comparison, showing performance changes across different models before and after `PRISM` enhancement.

**Long-term forecasting capability analysis.** After addressing the smoothing problem, our model effectively performs long-term forecasting by taking 1 input time step to predict the next 59. As the forecasting horizon extends, high-frequency information decays more slowly (He et al.). We employ the shallow water equations (Wu et al., 2024a) to describe wave dynamics near the equator, involving three variables:

Table 4: MSE reduction for scalar (wave height) and vector (U, V velocity) variables. The "Improvement" column shows the mean percentage decrease in MSE for each model after applying our method.

| MODEL | SCALAR MSE | VECTOR MSE | IMPROVEMENT |
|---|---|---|---|
| RESNET | 0.0782 ↓ 0.0701 | 0.0983 ↓ 0.0873 | 10.8% |
| SIMVP | 0.0345 ↓ 0.0298 | 0.0564 ↓ 0.0531 | 10.0% |
| FNO | 0.0986 ↓ 0.0763 | 0.1102 ↓ 0.0972 | 16.9% |
| EGNN | 0.0425 ↓ 0.0403 | 0.0609 ↓ 0.0576 | 6.1% |

water wave height and its corresponding U and V velocities. Table 4 shows that after integrating `PRISM`, all backbone networks yield lower mean squared errors (MSE) for both scalar (wave height) and vector (U, V) variables. Specifically, ResNet's scalar MSE decreases from 0.0782 to 0.0701, and vector MSE from 0.0983 to 0.0873. SimVP's scalar MSE drops from 0.0345 to 0.0298, and vector MSE from 0.0564 to 0.0531. FNO's scalar MSE falls from 0.0986 to 0.0763, and vector MSE from 0.1102 to 0.0972. EGNN's scalar MSE decreases from 0.0425 to 0.0403, and vector MSE from 0.0609 to 0.0576. These results confirm that `PRISM` significantly enhances forecasting accuracy.

As shown in Figure 4 (a), the visualization results indicate that EGNN+`PRISM` performs well in long-term forecasting (from step 1 to step 59), with predictions closely matching the ground truth. Next, as shown in Figure 4 (b), we convert the energy spectrum to a logarithmic scale to better illustrate the broad energy distribution. The results show that the energy spectrum of EGNN+`PRISM` aligns closely with the ground truth. Based on the original backbone network, this strongly validates that our method effectively enhances the model's long-term forecasting capability and improves its ability to learn both high- and low-frequency dynamics.

**Extreme event capability analysis.** As shown in Figure 4(c), we select SEVIR for analysis. The visualization results show that CNO produces smoother outputs, while extreme events are high-frequency, making analysis difficult. After combine `PRISM`, the model optimization shifts toward higher extreme event scores, resulting in more high-frequency outputs. This demonstrates that our method improves the model's ability to capture extreme events. The Figure 4(d) better reflects the improved consistency of our method in extreme events.

**Transfer learning capability analysis.** We first pretrain the model using `PRISM` on the source domain dataset, Prometheus-P, to achieve strong numerical forecasting accuracy and alignment with human preferences. Then, we transfer the model to two target datasets: Prometheus-T (with 20%, 40%, and 100% training data) and Rayleigh-Bénard (full dataset). This design evaluates performance changes when data is limited in fire scenarios and examines whether `PRISM` improves numerical accuracy and perceptual quality in a completely different convection process.

Table 5: **Transfer Learning Performance.** We pretrain on a source domain and transfer to two target datasets. For *Prometheus-T*, performance (MSE↓/SSIM↑) is evaluated with varying percentages of training data. For *Rayleigh-Bénard*, we compare training from scratch (FS) against fine-tuning with our method (+ PRISM), evaluated by MSE↓/MAPE↓.

| MODEL | TARGET DOMAIN 1: PROMETHEUS-T (FIRE SIMULATION) | | | | | | TARGET DOMAIN 2: RAYLEIGH-BÉNARD (CONVECTION) | | | |
|---|---|---|---|---|---|---|---|---|---|---|
| | DATA: 20% | | DATA: 40% | | DATA: 100% | | FROM SCRATCH (FS) | | + PRISM | |
| | MSE | SSIM | MSE | SSIM | MSE | SSIM | MSE | MAPE | MSE | MAPE |
| RESNET | 0.141 | 0.801 | 0.124 | 0.823 | 0.099 | 0.833 | 0.713 | 25.65 | 0.697 | 24.95 |
| U-NET | 0.157 | 0.793 | 0.115 | 0.838 | 0.106 | 0.831 | 0.127 | 13.73 | 0.114 | 12.73 |
| SIMVP | 0.121 | **0.901** | 0.102 | **0.928** | 0.034 | 0.924 | 0.093 | 20.01 | 0.088 | 14.83 |
| PASTNET | 0.112 | 0.891 | **0.071** | 0.923 | 0.031 | **0.941** | 0.114 | 11.84 | 0.113 | 11.09 |
| UNO | 0.132 | 0.821 | 0.118 | 0.865 | 0.050 | 0.895 | 0.112 | **11.60** | 0.107 | 10.30 |
| OURS | **0.106** | 0.891 | 0.079 | 0.913 | **0.029** | 0.911 | **0.103** | 12.31 | **0.098** | 10.29 |

Table 5 presents the key results. On Prometheus-T, as the training data increases from 20% to 100%, MSE decreases and SSIM improves. For example, ResNet's MSE drops from 0.141 to 0.099, and SSIM rises from 0.801 to 0.833. SimVP and PastNet also demonstrate high accuracy and stability. On Rayleigh-Bénard, compared to training from scratch, PRISM significantly reduces MSE and MAPE. For instance, ResNet's MSE decreases from 0.713 to 0.697, and MAPE drops from 25.647% to 24.953%. These results show that PRISM effectively enhances model accuracy and perceptual consistency in both fire spread and convection field forecasting, demonstrating strong transferability and practical value.

**Efficiency analysis.** As shown in Table 6, applying the PRISM framework slightly increases training time per epoch by about 10%-15% on average. For example, ResNet's training time rises from 18.200 seconds to 20.400 seconds. However, inference speed (FPS) experiences only a minor decrease, with ResNet dropping from 225.000 FPS to 220.000 FPS, indicating that PRISM has little impact on inference efficiency. Additionally, GPU memory usage increases slightly,

Table 6: **Efficiency Analysis.** We compare training time (s/epoch ↑), inference speed (FPS ↓), and peak GPU memory (GB ↑) before and after applying PRISM. Arrows indicate that changes represent computational overhead.

| MODEL | TRAIN (S/EPOCH) | INFER (FPS) | MEM (GB) |
|---|---|---|---|
| RESNET | 18.2 → 20.4 | 225 → 220 | 5.3 → 5.7 |
| U-NET | 25.3 → 27.8 | 180 → 174 | 6.1 → 6.4 |
| SIMVP | 23.9 → 25.6 | 210 → 205 | 6.9 → 7.2 |
| PASTNET | 31.5 → 34.1 | 160 → 157 | 7.8 → 8.1 |

with ResNet rising from 5.300 GB to 5.700 GB, mainly due to diversified sampling and the introduction of the preference model. Overall, PRISM enhances model performance while maintaining a moderate impact on computational efficiency and resource consumption, ensuring its feasibility and practicality in real-world applications.

## 5 CONCLUSION

In this paper, we introduce PRISM, a novel learning paradigm designed to bridge the fundamental gap between conventional numerical objectives and the physically grounded desiderata essential for dynamical system modeling. Traditional methods, while numerically optimized, often fail to capture the high-fidelity details and extreme events that are critical for scientific applications. Our framework innovatively integrates the principles of Direct Preference Optimization to address this limitation. By distilling the complex, often implicit, criteria of domain experts into a differentiable preference oracle, PRISM successfully transforms nuanced human priors into a tractable, end-to-end optimization signal. Extensive experiments across diverse benchmarks, including fluid dynamics, fire propagation, and extreme weather forecasting, provide compelling evidence of our approach's effectiveness and generalizability. As a plug-and-play module, PRISM consistently enhances the performance of various backbone architectures to capture high-frequency dynamics, predict rare events, and improve physical plausibility. This work not only validates a powerful new technique but, more importantly, charts a promising path toward a new generation of human-machine collaborative models for scientific discovery systems that learn not just to be numerically accurate, but to align with the profound intuition of human experts.

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

# THE USE OF LARGE LANGUAGE MODELS (LLMS)

LLMs were not involved in the research ideation or the writing of this paper.

# A  THEORETICAL CONVERGENCE ANALYSIS

In this section, we provide the theoretical analysis regarding the convergence of the PRISM framework. We formulate the joint optimization in Stage 3 as a composite optimization problem and prove its convergence to a stationary point. This theoretically grounds our claim that the bi-level optimization process converges to a stable equilibrium between numerical fidelity and human-aligned preference.

## A.1  PROBLEM FORMULATION

Recall the total objective function defined in Eq. (7) of the main paper. The optimization problem for the foundational model parameter $\theta$ is:

$$\min_{\theta} \mathcal{J}(\theta) = \mathcal{L}_{\text{MSE}}(\theta) + \lambda \mathcal{L}_{\text{DPO}}(\theta; \theta_{\text{ref}}) \tag{10}$$

where $\mathcal{L}_{\text{MSE}}(\theta) = \mathbb{E}_{(X,Y)\sim\mathcal{D}}[\|f_\theta(X) - Y\|^2]$ ensures numerical fidelity. The preference alignment term $\mathcal{L}_{\text{DPO}}$ is derived from the distilled Preference Oracle $S_\phi$.

**Remark on Differentiability:** While raw physical metrics (e.g., Critical Success Index, Hit Rate) are often non-differentiable or discrete, our Preference Oracle $S_\phi$ is parameterized by a neural network with smooth activation functions (e.g., Sigmoid, GELU). Consequently, the distilled preference signal transforms the non-smooth metric optimization into a smooth surrogate objective $\mathcal{J}(\theta)$, enabling gradient-based optimization.

## A.2  ASSUMPTIONS

To establish the convergence of the stochastic gradient descent (SGD) algorithm used in PRISM, we make the following standard assumptions for non-convex optimization problems:

- **Assumption 1 (Lower Bound):** The objective function $\mathcal{J}(\theta)$ is bounded from below. That is, there exists a constant $\mathcal{J}^* > -\infty$ such that $\mathcal{J}(\theta) \geq \mathcal{J}^*$ for all $\theta \in \mathbb{R}^d$.

- **Assumption 2 (L-Smoothness):** The objective function $\mathcal{J}(\theta)$ is $L$-smooth. This implies that $\mathcal{J}$ is differentiable and its gradient is $L$-Lipschitz continuous:

$$\|\nabla\mathcal{J}(\theta_1) - \nabla\mathcal{J}(\theta_2)\| \leq L\|\theta_1 - \theta_2\|, \quad \forall \theta_1, \theta_2 \in \mathbb{R}^d \tag{11}$$

  Given that both the predictive model $f_\theta$ and the preference model $S_\phi$ are deep neural networks composed of smooth operators, this assumption generally holds locally.

- **Assumption 3 (Bounded Variance):** The stochastic gradients $g_t$ computed on mini-batches are unbiased estimates of the full gradient, and their variance is bounded by $\sigma^2$:

$$\mathbb{E}[g_t] = \nabla\mathcal{J}(\theta_t), \quad \mathbb{E}[\|g_t - \nabla\mathcal{J}(\theta_t)\|^2] \leq \sigma^2 \tag{12}$$

## A.3  CONVERGENCE THEOREM

We define a **Pareto Stationary Point** as a solution where the gradient of the composite loss vanishes, representing a stable trade-off between the physics-based loss and the preference-based reward.

**Theorem 1** (**Convergence to Stationary Point**). *Let $\{\theta_t\}_{t=0}^{T-1}$ be the sequence of parameters generated by SGD with a constant learning rate $\eta$ satisfying $0 < \eta \leq \frac{1}{L}$. Under Assumptions 1-3, the algorithm converges to a stationary point in expectation. Specifically, the average squared norm of the gradients satisfies:*

$$\mathbb{E}\left[\frac{1}{T}\sum_{t=0}^{T-1}\|\nabla\mathcal{J}(\theta_t)\|^2\right] \leq \frac{2(\mathcal{J}(\theta_0) - \mathcal{J}^*)}{T\eta} + L\eta\sigma^2 \tag{13}$$

*As $T \to \infty$, if we choose a decaying learning rate, the gradient norm converges to zero.*

*Proof.* Based on the $L$-smoothness assumption, we have the following quadratic upper bound inequality:

$$\mathcal{J}(\theta_{t+1}) \leq \mathcal{J}(\theta_t) + \langle \nabla \mathcal{J}(\theta_t), \theta_{t+1} - \theta_t \rangle + \frac{L}{2} \|\theta_{t+1} - \theta_t\|^2 \tag{14}$$

Substituting the SGD update rule $\theta_{t+1} = \theta_t - \eta g_t$:

$$\mathcal{J}(\theta_{t+1}) \leq \mathcal{J}(\theta_t) - \eta \langle \nabla \mathcal{J}(\theta_t), g_t \rangle + \frac{L\eta^2}{2} \|g_t\|^2 \tag{15}$$

Taking the expectation with respect to the stochasticity at step $t$:

$$\mathbb{E}[\mathcal{J}(\theta_{t+1})] \leq \mathcal{J}(\theta_t) - \eta \|\nabla \mathcal{J}(\theta_t)\|^2 + \frac{L\eta^2}{2} (\mathbb{E}\|g_t - \nabla \mathcal{J}(\theta_t)\|^2 + \|\nabla \mathcal{J}(\theta_t)\|^2) \tag{16}$$

$$\leq \mathcal{J}(\theta_t) - \left( \eta - \frac{L\eta^2}{2} \right) \|\nabla \mathcal{J}(\theta_t)\|^2 + \frac{L\eta^2\sigma^2}{2} \tag{17}$$

Assuming $\eta \leq \frac{1}{L}$, we have $\eta - \frac{L\eta^2}{2} \geq \frac{\eta}{2}$. Rearranging the terms gives:

$$\frac{\eta}{2} \|\nabla \mathcal{J}(\theta_t)\|^2 \leq \mathcal{J}(\theta_t) - \mathbb{E}[\mathcal{J}(\theta_{t+1})] + \frac{L\eta^2\sigma^2}{2} \tag{18}$$

Summing over $t = 0$ to $T - 1$ and taking total expectation:

$$\sum_{t=0}^{T-1} \frac{\eta}{2} \mathbb{E}\|\nabla \mathcal{J}(\theta_t)\|^2 \leq \mathcal{J}(\theta_0) - \mathbb{E}[\mathcal{J}(\theta_T)] + \frac{TL\eta^2\sigma^2}{2} \tag{19}$$

Since $\mathcal{J}(\theta_T) \geq \mathcal{J}^*$, we have $\mathcal{J}(\theta_0) - \mathbb{E}[\mathcal{J}(\theta_T)] \leq \mathcal{J}(\theta_0) - \mathcal{J}^*$. Dividing by $\frac{T\eta}{2}$, we obtain the theorem result. $\square$

### A.4 DISCUSSION ON EQUILIBRIUM

The derived stationary point $\theta^*$ satisfies $\nabla \mathcal{J}(\theta^*) = \nabla \mathcal{L}_{\text{MSE}}(\theta^*) + \lambda \nabla \mathcal{L}_{\text{DPO}}(\theta^*) = 0$. This implies:

$$\nabla \mathcal{L}_{\text{MSE}}(\theta^*) = -\lambda \nabla \mathcal{L}_{\text{DPO}}(\theta^*) \tag{20}$$

This condition characterizes a **Pareto Equilibrium**: at this state, any infinitesimal update to improve the preference score $\mathcal{L}_{\text{DPO}}$ would result in a strictly opposing degradation in numerical fidelity $\mathcal{L}_{\text{MSE}}$, weighted by $\lambda$. This theoretical result validates that PRISM effectively converts the discrete, expert-guided preference alignment problem into a stable, differentiable optimization task.

## B MODEL SUMMARY

The pseudo-algorithm of `PRISM` is shown in Algorithm 1.

**Algorithm 1** Overview of the Method

**Require:** Training dataset $\mathcal{D}$; Perturbation intensity $\sigma$; Balancing coefficient $\lambda$; Learning rate $\eta$.

1: Initialize foundation model parameters $\Theta$ and preference model parameters $\phi$.
2: // -- Stage 1: Pretrain Foundation Model --
3: **for** number of pretraining epochs **do**
4:     Sample a mini-batch $(X, Y_{\text{true}})$ from $\mathcal{D}$.
5:     Compute prediction: $\hat{Y} \leftarrow \text{FoundationModel}_\Theta(X)$.
6:     Calculate MSE loss: $\mathcal{L}_{\text{MSE}} \leftarrow \|\hat{Y} - Y_{\text{true}}\|_2^2$.
7:     Update foundation model parameters $\Theta$ using gradient descent on $\mathcal{L}_{\text{MSE}}$.
8: **end for**
9: -- Stage 2: Diverse Sample Generation --
10: Initialize preference dataset $\mathcal{D}_{\text{pref}} \leftarrow \emptyset$.
11: **for** each input $X$ in $\mathcal{D}$ **do**
12:     Generate a set of diverse predictions $\{\hat{Y}_i\}_{i=1}^N$ from $X$ using the pretrained model $\Theta$.
13:     Filter and form preference pairs $(Y_j^+, Y_j^-)$ from $\{\hat{Y}_i\}$ using physical metrics $M(\cdot)$.
14:     Add generated pairs to $\mathcal{D}_{\text{pref}}$.
15: **end for**
16: -- Stage 3: Train Preference Model --
17: **for** number of preference training epochs **do**
18:     Sample a preference pair $(Y_j^+, Y_j^-)$ from $\mathcal{D}_{\text{pref}}$.
19:     Compute preference loss: $\mathcal{L}_{\text{Pref}} \leftarrow -\log\left(\sigma\left(S_\phi(Y_j^+) - S_\phi(Y_j^-)\right)\right)$.
20:     Update preference model parameters $\phi$ using gradient descent on $\mathcal{L}_{\text{Pref}}$.
21: **end for**
22: -- Stage 4: Joint Optimization --
23: **for** number of joint optimization epochs **do**
24:     Sample a mini-batch $(X, Y_{\text{true}})$ from $\mathcal{D}$.
25:                                                ▷ Compute base task loss
26:     $\mathcal{L}_{\text{MSE}} \leftarrow \mathbb{E}\|\text{FoundationModel}_\Theta(X) - Y_{\text{true}}\|_2^2$.
27:                   ▷ Compute preference alignment loss using the fixed preference model
28:     Generate diverse outputs $\{Y_k\}$ and form pairs $(Y_k^+, Y_k^-)$.
29:     $\mathcal{L}_{\text{Align}} \leftarrow \mathbb{E}_{(Y^+, Y^-)}\left[\log\left(1 + e^{S_\phi(Y^-) - S_\phi(Y^+)}\right)\right]$.
30:                       ▷ Combine losses and update foundation model
31:     Total loss: $\mathcal{L}_{\text{Total}} \leftarrow \mathcal{L}_{\text{MSE}} + \lambda \mathcal{L}_{\text{Align}}$.
32:     Update foundation model parameters $\Theta \leftarrow \Theta - \eta \nabla_\Theta \mathcal{L}_{\text{Total}}$.
33: **end for**
34: **return** Optimized foundation model parameters $\Theta^*$.

## C   EVALUATION METRICS

To comprehensively evaluate the performance of our proposed `PRISM` framework, we employ a set of evaluation metrics tailored to different aspects of dynamic system prediction. These metrics include Mean Squared Error (MSE), Structural Similarity Index (SSIM), Critical Success Index (CSI), and Mean Percentage Absolute Error (MPAE). Below, we provide the mathematical formulations and detailed descriptions of each metric.

### C.1   MEAN SQUARED ERROR (MSE)

Mean Squared Error quantifies the average squared difference between the predicted values and the ground truth. It is a fundamental metric for assessing numerical accuracy in predictions.

$$\text{MSE} = \frac{1}{N} \sum_{i=1}^{N} \left( Y_{\text{pred},i} - Y_{\text{true},i} \right)^2 . \tag{21}$$

**Description:** MSE measures the average of the squares of the errors between predicted values ($Y_{\text{pred}}$) and true values ($Y_{\text{true}}$). A lower MSE indicates higher predictive accuracy, making it a crucial metric for evaluating numerical consistency in dynamic system modeling.

### C.2   STRUCTURAL SIMILARITY INDEX (SSIM)

Structural Similarity Index assesses the similarity between two images by considering luminance, contrast, and structural information. It is particularly useful for evaluating visual perceptual quality.

$$\text{SSIM}(x, y) = \frac{(2\mu_x\mu_y + C_1)(2\sigma_{xy} + C_2)}{(\mu_x^2 + \mu_y^2 + C_1)(\sigma_x^2 + \sigma_y^2 + C_2)}. \tag{22}$$

**Description:** SSIM evaluates the similarity between two images ($x$ and $y$) by analyzing their luminance ($\mu_x$, $\mu_y$), contrast ($\sigma_x$, $\sigma_y$), and structural correlation ($\sigma_{xy}$). Constants $C_1$ and $C_2$ stabilize the division to prevent instability when the denominators are close to zero. SSIM values range from -1 to 1, where higher values indicate greater structural similarity and better visual quality of the predictions.

### C.3   CRITICAL SUCCESS INDEX (CSI)

Critical Success Index measures the accuracy of predicting extreme events by evaluating the proportion of correctly predicted events against the total number of observed and predicted events.

$$\text{CSI} = \frac{\text{Hits}}{\text{Hits} + \text{Misses} + \text{False Alarms}}. \tag{23}$$

**Description:** CSI assesses the model's capability to accurately predict extreme events, such as severe weather phenomena.

- Hits: Correctly predicted extreme events. - Misses: Actual extreme events that were not predicted. - False Alarms: Predicted extreme events that did not occur.

A higher CSI indicates better performance in identifying and predicting extreme events, which are often rare but critical for applications like disaster prevention and resource management.

### C.4   MEAN PERCENTAGE ABSOLUTE ERROR (MPAE)

Mean Percentage Absolute Error quantifies the average absolute percentage difference between the predicted values and the ground truth. It is particularly useful for assessing relative errors in physical metrics.

$$\text{MPAE} = \frac{1}{N} \sum_{i=1}^{N} \left| \frac{Y_{\text{pred},i} - Y_{\text{true},i}}{Y_{\text{true},i}} \right| \times 100\%. \tag{24}$$

**Description:** MPAE measures the average absolute percentage error between predicted values ($Y_{\text{pred}}$) and true values ($Y_{\text{true}}$). In the context of the Rayleigh-Bénard Convection (RBC) dataset, we compute

MPAE for the turbulent kinetic energy spectrum derived from the U and V velocity components. This approach effectively captures the relative errors in the energy distribution across different scales, which is essential for evaluating the physical consistency and accuracy of turbulence modeling. Since the turbulent kinetic energy spectrum involves complex interactions between velocity components, directly measuring errors in U and V velocities may not adequately reflect the model's performance in capturing the underlying physical phenomena. Therefore, MPAE serves as a more informative metric for assessing the quality of predictions in such scenarios.

### C.5 MEAN ABSOLUTE ERROR (MAE)

Mean Absolute Error provides a straightforward measure of prediction accuracy by averaging the absolute differences between predicted values and true values.

$$\text{MAE} = \frac{1}{N} \sum_{i=1}^{N} \left| Y_{\text{pred},i} - Y_{\text{true},i} \right| . \tag{25}$$

**Description:** MAE calculates the average absolute deviation of the predictions from the actual values. Unlike MSE, MAE does not square the errors, making it less sensitive to outliers and providing a more interpretable measure of average error magnitude. Lower MAE values indicate better predictive performance.

### C.6 APPLICATION OF METRICS IN DATASETS

Different datasets and prediction tasks emphasize various aspects of model performance, necessitating the selection of appropriate evaluation metrics:

- Prometheus Dataset: We focus on both numerical accuracy and visual perceptual quality, utilizing MSE and SSIM to assess the fidelity and structural similarity of the predictions.

- Rayleigh-Bénard Convection (RBC) Dataset: Given the complexity of turbulent energy distributions, we employ MPAE to evaluate the relative errors in the turbulent kinetic energy spectrum derived from U and V velocity components. This choice ensures that the physical consistency and energy distribution are accurately captured by the model.

- SEVIR Dataset: For extreme weather event prediction, we use MSE to measure numerical accuracy and CSI to evaluate the model's ability to correctly identify and predict extreme events.

By integrating these metrics, we ensure a holistic evaluation of `PRISM`, capturing both quantitative accuracy and qualitative aspects aligned with human perceptual and physical consistency requirements.

## D RELATED WORK

**Dynamical System Modeling.** Early dynamic system prediction relied on numerical simulations or closed-form PDE models for analytical system evolution. With deep learning advances, many studies now use end-to-end neural networks optimized with metrics like MSE (Li et al., 2020; Xiong et al., 2024). Typically employing CNNs (Wu et al., 2024e; 2023) or ST architectures (Gao et al., 2022a), these models extract essential spatiotemporal correlations to minimize prediction errors. While they reduce average errors and speed up predictions, their focus on overall distributions hampers the accurate modeling of extreme or sudden dynamics and often lacks physical consistency and interpretability.

**Generative and Diverse Sampling.** To capture higher-order uncertainties beyond average errors, researchers utilize generative models like GANs, VAEs, and diffusion models (Zhang et al., 2023; Li et al., 2024; Fotiadis et al., 2023). GANs perform adversarial training between real and predicted distributions, VAEs estimate explicit probabilities, and diffusion models characterize complex distributions via perturbation and generation processes. Additionally, diverse sampling techniques such as input perturbations, multimodal fusion, and discrete embedding replacements generate richer predictions (Bi et al., 2022; Wu et al.). However, challenges like training stability, computational costs, and incorporating human preferences remain.

**Physical and Preference Fusion Methods.** Recent work embeds physical constraints or integrates human prior knowledge into deep models. Incorporating physical laws into network structures or post-processing reduces deviations from real processes, enhancing reliability (Zhang et al., 2023; Rao et al., 2023; Raissi et al., 2019). Additionally, human preference learning introduces expert scores (Rafailov et al., 2024; Chen et al.; Zhou et al., 2024), annotations, or quality assessments into predictions by using preference models to translate subjective evaluations into learnable objectives. These fusion methods preserve accuracy, align with application needs and human perception, and enhance the detection of high-risk scenarios while improving interpretability and trustworthiness.

## E    DETAILS OF COMPARED APPROACHES

The compared approaches involved in this study is as follows:

- **ResNet** He et al. (2016) introduces residual blocks to solve the degradation problem in deep networks. It allows the network to be deeper and easier to train by using skip connections to directly pass information.

- **U-Net** Ronneberger et al. (2015) is a convolutional neural network initially used for biomedical image segmentation. It has a symmetric U-shaped structure and uses skip connections to link the encoder and decoder, enabling efficient feature fusion.

- **ViT** Dosovitskiy et al. (2021) applies the Transformer model to image recognition. It divides the image sample into patches and uses self-attention mechanisms to process these patches, balancing computational efficiency and performance.

- **SwinT** Liu et al. (2021) introduces a sliding window mechanism for effective local and global feature extraction. It is suitable for various computer vision tasks.

- **SimVP** Gao et al. (2022a) is a straightforward video prediction framework that utilizes a simple convolutional architecture to model spatiotemporal dependencies in dynamic systems. By minimizing architectural complexity, SimVP achieves competitive performance with reduced computational overhead, making it effective for various forecasting tasks in scientific computing and engineering applications.

- **PastNet** Wu et al. (2024e) is a spatiotemporal predictive network that leverages physical information to enhance video prediction accuracy. PastNet integrates past states through a time-aware state transition mechanism, allowing the model to capture complex temporal dynamics and improve stability in long-term predictions, particularly in scenarios involving fluid dynamics and weather forecasting.

- **FNO** Li et al. (2020) uses Fourier transforms for global feature extraction, suitable for processing continuous field data and efficiently solving PDEs.

- **UNO** Ashiqur Rahman et al. (2022) combines the U-Net architecture with optimization methods to enhance feature extraction and fusion capabilities, improving model performance.

- **CNO** Raonic et al. (2024) combines convolution operations with operator learning, focusing on high-dimensional continuous data and modeling complex dynamic systems.

- **MGN** Pfaff et al. (2020) employs multiple graph neural network layers to effectively capture intricate relationships and interactions within dynamic systems. By utilizing multi-scale graph representations, MGN models the complex dependencies inherent in scientific computing tasks, thereby enhancing prediction accuracy and robustness across diverse applications such as turbulence simulation and climate modeling.

- **EGNN** Satorras et al. (2021) (Equivariant Graph Neural Network) is designed to preserve geometric symmetries in data, making it highly suitable for physical simulations and dynamical system modeling. EGNN ensures equivariance with respect to rotations and translations, maintaining the physical consistency of predictions. This characteristic improves the reliability and interpretability of model outputs, particularly in applications involving rigid body dynamics and molecular simulations.

