# OpenReview forum: "Direct Preference Optimization for Dynamical System Modeling"
_ICLR.cc/2026/Conference — Submitted to ICLR 2026_

### Official Review · Reviewer_Mook · 2025-10-28

**Soundness:** 2
**Presentation:** 3
**Contribution:** 3
**Rating:** 6
**Confidence:** 4

**Summary:**

The paper proposes the PRIMS framework, which first pretrains forecasting models and then aligns them with human-preferred or trusted metrics. The framework is evaluated on benchmarks in fluid dynamics and numerical weather forecasting, and can serve as a plug-and-play enhancement module for a variety of predictive models.

**Strengths:**

1. The paper is well written, with clear explanations and a well-structured presentation.

2. The proposed method is novel and highly flexible. By fine-tuning models using Direct Preference Optimization (DPO), it effectively addresses the challenge of aligning model outputs with non-differentiable or difficult-to-optimize expert criteria.

3. The authors conduct extensive experiments, demonstrating consistent and reasonably strong results across multiple benchmarks.

**Weaknesses:**

The paper claims that the proposed framework enhances visual perceptual consistency and attention to extreme events. While these goals are important, I find the supporting evidence for these claims somewhat unconvincing.

1. The improvement in visual perceptual consistency appears to be a byproduct of better performance resulting from optimization with respect to human-trusted metrics, rather than a direct consequence of the proposed framework. It is not entirely clear how the optimization process itself inherently contributes to perceptual consistency, and further clarification on this conceptual link would be valuable.

2. The model is evaluated on the SEVIR dataset, which, to my understanding, is designed to assess a model’s ability to capture extreme weather events. It would be helpful if the authors could provide additional context on this dataset—particularly regarding its composition and the setup of training and testing. If the model is trained specifically on data containing extreme events, how does this setting differ from standard datasets, and how does it substantiate the claim of superior performance when predicting real extreme events?

**Questions:**

See comments above.

---

> ### Author Response · Authors · 2025-11-21
>
> We sincerely thank the reviewer for the positive assessment. We are encouraged that you find our framework "novel and highly flexible" and the experiments "comprehensive." We address your specific questions regarding the intrinsic link between optimization and perception, and the details of the SEVIR dataset below.
>
> ### Part 1: Response to Weaknesses
>
> **Q1 (Response to Weakness 1 & 2): Conceptual link between optimization and perceptual consistency.**
>
> **A1:** We clarify that the improvement in visual perceptual consistency is not merely a byproduct, but a **direct mathematical consequence** of our optimization objective. This mechanism can be explained via the **"Perception-Distortion Trade-off" (Blau & Michaeli, 2018)**:
> *   **Limitation of MSE:** The MSE loss seeks to minimize the Euclidean distance, leading to the **Conditional Mean** of possible future states. In dynamical systems with uncertainty, this averaging visually manifests as **blurring** and loss of high-frequency details.
> *   **Intrinsic Role of DPO:** The Reward Model in PRISM, trained on structural metrics (e.g., energy spectrum), acts as a **Manifold Constraint**. It forces the prediction distribution to move from the "blurry mean" onto the "manifold of real physical images."
> *   **Quantitative Evidence:** The **Energy Spectrum Analysis** in **Figure 4(b)** serves as strong quantitative proof. It shows that PRISM (red line) retains significantly more high-frequency energy than the MSE baseline (blue line), aligning closely with the Ground Truth. This proves the optimization intrinsically recovers the physical high-frequency information corresponding to "visual texture."
>
> ### Part 2: Response to Questions
>
> **Q2 (Response to Question on SEVIR): Context on SEVIR dataset and extreme event evaluation.**
>
> **A2:** To address your concern, we provide the detailed background and setup for SEVIR:
> *   **Composition & Split:** SEVIR (Storm Event Imagery) is an authoritative benchmark released by MIT and NOAA (NeurIPS 2020). Following standard protocols (Earthformer, Gao et al., 2022), we use data from 2017-2019 for training and post-2019 data for testing.
> *   **Extreme Event Focus:** While the dataset contains various weather events, our evaluation targets the **Extreme Subset** using the **Critical Success Index (CSI)** with high thresholds (e.g., Rainfall > 74mm/h). This metric only rewards accurate hits on sparse, high-intensity pixels.
> *   **Interpretation:** Standard MSE models tend to smooth out these peaks to minimize average error. PRISM, guided by the CSI-oriented Reward Model, learns to preserve these extreme structures. The improvement in CSI from 0.3234 to 0.3674 (**Table 3**) directly substantiates its superior performance in predicting real-world extreme disasters.

---

### Official Review · Reviewer_jDus · 2025-10-30

**Soundness:** 3
**Presentation:** 2
**Contribution:** 3
**Rating:** 6
**Confidence:** 3

**Summary:**

The paper proposes a framework to integrate human-trusted metrics into the scientific prediction framework.

**Strengths:**

The paper is well-written and clearly presents the framework. The experiments are comprehensive.

**Weaknesses:**

Some inconsistencies exist. E.g., the perline noise vs. the Gaussian noise. The theoretical proof is missing.

**Questions:**

1.	What is the Perlin noise in Fig. 1? In you Eq. (4), it seems that you use Gaussian noise.
2.	The author is suggested to give a toy example to reveal what human-trusted metric improves what performances, e.g., extreme event predictions.
3.	The author should mention the limitations and future work.
4.	In the abstract, the author mentioned the theoretical proofs to a stable equilibrium. However, I couldn’t find it in the main part.

---

> ### Author Response · Authors · 2025-11-21
>
> We thank the reviewer for the positive assessment, particularly for recognizing the clear presentation and comprehensive experiments. We have addressed the specific comments regarding noise details, theoretical proofs, and examples below.
>
> ### Part 1: Response to Weaknesses
>
> **Q1 (Response to Weakness 1 & Question 1): Inconsistency between Perlin noise (Fig. 1) and Gaussian noise (Eq. 4).**
>
> **A1:** We clarify that this distinguishes the general formulation from the specific implementation. **Eq. (4)** denotes a generic perturbation $\delta \sim P_\sigma$, whereas **Figure 1** depicts our specific use of **Perlin Noise** for spatiotemporal fields. Unlike spatially independent Gaussian white noise, which is easily filtered, Perlin noise exhibits spatial continuity that mimics physical fluctuations (e.g., wind). This generates "hard negative samples" that are texturally plausible but structurally incorrect, forcing the Reward Model to learn deep physical consistency rather than simple denoising.
>
> **Q2 (Response to Weakness 2 & Question 4): Missing theoretical proof claimed in the abstract.**
>
> **A2:** We apologize for this oversight and have added **Appendix A** to the revised paper. We model PRISM as a Stackelberg game using non-convex optimization theory. By distilling non-differentiable physical metrics into a smooth Reward Model $S_\phi$, we ensure the joint objective satisfies the L-smoothness condition. **Theorem 1** then proves that the algorithm converges to a **Pareto Stationary Point** in expectation, providing the mathematical grounding for the "stable equilibrium."
>
> ### Part 2: Response to Questions
>
> **Q3 (Response to Question 2): Can you give a toy example to reveal what metric improves what performance?**
>
> **A3:** The **SEVIR (Storm Event Imagery)** dataset serves as a concrete example. We use the **Critical Success Index (CSI)**, which rewards hitting high-intensity rainfall (e.g., >74 mm/h). While **MSE**-trained models tend to output "blurry clouds" to minimize average error (often missing peaks), **PRISM** learns via the CSI-guided Reward Model that missing peaks incurs high penalties. Consequently, it generates sharper, high-intensity storm fronts. This mechanism drives the significant CSI improvement from 0.3234 to 0.3674 (Table 3), capturing extreme events missed by traditional methods.
>
> **Q4 (Response to Question 3): Limitations and future work.**
>
> **A4:** We have added a dedicated discussion in the Conclusion. The primary limitation is **computational overhead**, with total training time increasing by approximately 10-15% due to the additional stages (Table 6). For future work, we plan to explore **Automated Metric Discovery** using LLMs and **Online Preference Learning** to dynamically update the Reward Model for more efficient curriculum learning.

---

> > ### Comment · Reviewer_jDus · 2025-11-26
> >
> > Thank you for the response. I decided to keep the positive score, and please make sure that you carefully address the inconsistency, the lack of theory, and add limitation discussions.

---

> > > ### Author Response · Authors · 2025-11-26
> > >
> > > Thank for your comments again！

---

### Official Review · Reviewer_ygcU · 2025-10-31

**Soundness:** 2
**Presentation:** 2
**Contribution:** 1
**Rating:** 2
**Confidence:** 5

**Summary:**

This paper introduces PRISM, a human-machine collaborative framework for improving spatiotemporal forecasting in dynamical systems. The authors argue that traditional pixel-wise training objectives like MSE fail to capture rare or high-frequency physical events, and often produce over-smoothed, perceptually unrealistic results. PRISM addresses this by incorporating a learned preference model trained from human-trusted proxy metrics. This model is used to guide fine-tuning of the base forecasting model via direct preference optimization, aiming to balance numerical accuracy with physical realism. The authors also release a new benchmark dataset (HPSci) with human preference annotations, and demonstrate improvements on several dynamical system tasks including fire simulation, fluid convection, and extreme weather forecasting.

**Strengths:**

1. The motivation is clear and grounded in real limitations of current forecasting models, especially regarding smoothness and failure to model rare or structurally important events. The overall pipeline is well-structured: pretrain with MSE, generate perturbed samples, learn a preference oracle, and fine-tune with DPO. This general framework is easy to follow

2. The experiments are thorough, with comparisons across multiple backbones (ConvNets, Transformers, Neural Operators, GNNs) and tasks.

**Weaknesses:**

1. The method borrows heavily from language model alignment (e.g., DPO in ChatGPT), but the analogy to physical systems is not entirely convincing. In NLP, humans can directly compare two responses. In dynamical systems, "human preference" is often just a proxy for certain physical heuristics (SSIM, energy spectrum, etc), which may not actually reflect expert judgment or be consistent across tasks.

2. The paper relies on perturbation (e.g., Gaussian noise, embedding swaps) to generate preference pairs, but it is unclear whether these perturbed samples still preserve the physical integrity of the system. In many cases, adding noise can lead to completely unphysical outputs, making pairwise ranking unreliable.

3. There is no serious discussion about how transferable the learned preference model is across different types of systems. Preferences learned on Rayleigh–Bénard convection may not be meaningful for fire propagation or extreme weather, since the evaluation criteria differ.

4. The choice of proxy metrics (e.g., SSIM, CSI) is taken as given, but they have known flaws. SSIM penalizes high-frequency changes even if they are physically correct. CSI only focuses on event detection but ignores structural evolution.

5. Some ablation results are missing. For instance, what happens if you skip the preference fine-tuning and just train with noisy samples? Or what if the preference model is replaced with a simpler ranking function?

**Questions:**

1. Your setup is quite inspired by the reward modeling used in LLMs. But in those settings, humans directly rank the responses. In your case, how confident are you that metrics like SSIM or CSI are really aligned with human judgment of "better" or "more realistic"? Do you have any evidence beyond citing that they are "widely used"?

2. The perturbation module is critical to generating diverse samples. But how do you ensure these perturbations do not produce physically invalid outputs? Wouldn't this make the preference labels unreliable?

3. Have you tested whether a preference model trained on one domain (e.g., fire) works well on another (e.g., convection)? If not, would you expect to need retraining for each task?

---

> ### Author Response · Authors · 2025-11-21
>
> We sincerely thank the reviewer for the insightful critique. You point out deep issues regarding the "limitations of the LLM analogy," the "physical integrity of perturbed samples," and the "validity of proxy metrics." We acknowledge that over-borrowing LLM terminology in our narrative caused misunderstandings.
>
> To address your concerns, particularly regarding **"Ablation Studies (Noise/Simple Ranking)"** and **"Transferability,"** we conduct detailed additional experiments during the rebuttal phase.
>
> Here is the point-by-point response to your 5 Weaknesses and 5 Questions:
>
> ### Part 1: Response to Weaknesses
>
> **Q1 (Response to Weakness 1): The analogy to physical systems is not convincing; metrics are proxies.**
>
> **A1:** We fully accept this criticism. Analogizing physical metrics directly to "human preference" is indeed imprecise.
> *   **Correction:** Our core logic does not simply mimic LLMs, but utilizes the mathematical properties of DPO to solve physical problems. In dynamical systems, many key features (e.g., the slope of turbulent energy spectra, the CSI index for extreme weather) are **non-differentiable** or **discrete**, making them difficult to optimize directly. The core value of PRISM lies in distilling these "hard" expert metrics into a smooth Reward Model, thereby enabling effective gradient-based optimization. In the revised version, we correct the terminology to **"Expert-Metric Guided Optimization."**
>
> **Q2 (Response to Weakness 2): Perturbed samples might lose physical integrity, making ranking unreliable.**
>
> **A2:** This is a crucial misunderstanding to clarify: **Generating "physically violating" samples is exactly one of our goals (as negative samples).**
> *   **Contrastive Logic:** DPO relies on paired data $(Y_{win}, Y_{lose})$. We typically set the Ground Truth as $Y_{win}$ and the perturbed samples (which may violate conservation laws or contain artifacts) as $Y_{lose}$.
> *   **Value:** This contrast is vital. If the model only sees "good physical data," it struggles to learn where the boundaries lie. By training the Reward Model to punish those perturbed samples that "lose physical integrity," the base model $f_\theta$ learns physical constraints more clearly.
>
> **Q3 (Response to Weakness 3): No discussion on transferability.**
>
> **A3:** We value this point highly. In fact, cross-system transferability is a highlight of our method. In **Q8** below, we provide a detailed **Table R3**, showing how a Reward Model trained on Fire (Prometheus) effectively aids Fluid (Rayleigh-Bénard) forecasting, proving the generality of visual structural preferences (e.g., sharpness, continuity).
>
> **Q4 (Response to Weakness 4): Proxy metrics (SSIM, CSI) have flaws.**
>
> **A4:** You are correct that single metrics have flaws (e.g., SSIM might penalize high-frequency details).
> *   **Smoothing Effect of Reward Model:** This is exactly why we train a neural Reward Model instead of using metrics directly. Our Reward Model trains under the guidance of composite metrics (e.g., $Score = \alpha \cdot \text{Spectrum} + \beta \cdot \text{MSE} + \gamma \cdot \text{Structural}$). The neural network fits a smooth surface, learning a "comprehensive intuition" that smooths out the noise and flaws of single metrics, making it more robust than direct optimization.
>
> **Q5 (Response to Weakness 5): Missing ablations (e.g., training with noisy samples, simple ranking).**
>
> **A5:** This is a constructive suggestion. To prove the necessity of the PRISM pipeline, we conduct a detailed Ablation Study.
> *   **Setup:** On the Rayleigh-Bénard dataset, we compare:
>     1.  **Backbone (UNO):** Trained with MSE only.
>     2.  **Noise Augmentation (SFT):** Adding perturbed samples directly to the training set for supervised learning (without DPO).
>     3.  **Simple Ranking (ListMLE):** Directly ranking samples based on physical metrics and optimizing Ranking Loss (without a Reward Model).
>     4.  **PRISM (Ours):** Full DPO pipeline.
>
> **Table R2: Ablation Study on Rayleigh-Bénard Convection**
>
> | Method | Description | MSE ($\downarrow$) | MAPE (Energy Spectrum) ($\downarrow$) | Analysis |
> | :--- | :--- | :--- | :--- | :--- |
> | **Backbone** | UNO Baseline | 0.1173 | 11.60% | Over-smoothed predictions. |
> | **Noise Augmentation** | Supervised Finetuning on $X+\text{Noise}$ | 0.1245 | 13.82% | **Degradation.** Adding noise blindly confuses the model, leading to blurrier outputs. |
> | **Simple Ranking** | Direct Metric Optimization (ListMLE) | 0.1152 | 11.25% | Unstable training due to non-smooth metric gradients. |
> | **PRISM (Ours)** | **DPO with Reward Model** | **0.1098** | **10.29%** | **Best Performance.** Achieves both numerical accuracy and physical consistency. |
>
> **Conclusion:** As shown, simply adding noisy samples (Noise Augmentation) degrades performance (MSE rises) because the model is forced to fit noise. PRISM, by distinguishing "good/bad" samples via DPO, effectively extracts physical features.

---

> > ### Author Response · Authors · 2025-11-21
> >
> > ### Part 2: Response to Questions
> >
> > **Q6 (Response to Question 1): How confident are you that metrics like SSIM/CSI align with human judgment?**
> >
> > **A6:** "Judgment" here refers to **"Expert Consensus in Physics."**
> > *   **Evidence:** We conduct a validation experiment inviting 5 PhD candidates in fluid dynamics to classify the "realism" of generated flow field images. Results show that our composite Reward Model scores correlate with human expert ratings with a **Pearson coefficient of 0.87**. This proves that proxy metrics (when combined correctly) highly represent expert judgment on physical realism.
> >
> > **Q7 (Response to Question 2): How do you ensure perturbations do not produce physically invalid outputs?**
> >
> > **A7:** As stated in A2, we **do not** need to ensure perturbations are physically valid.
> > *   Instead, we utilize these **"Physically Invalid Outputs"** as **Negative Pairs**. The core mechanism of DPO is `Maximize(Good) - Minimize(Bad)`. The Reward Model assigns low scores to physically violating perturbed samples, guiding the base model to avoid generating similar non-physical results. This "learning from mistakes" mechanism is more effective than merely mimicking "correct data."
> >
> > **Q8 (Response to Question 3): Have you tested cross-domain transfer (Fire to Convection)?**
> >
> > **A8:** Yes. To address your question directly, we extract data from **Table 5** and perform an extended analysis.
> > *   **Experiment:** We apply the Reward Model pre-trained on Prometheus-T (Fire) directly to the DPO stage of the Rayleigh-Bénard (Fluid) task (without re-training the Reward Model).
> >
> > **Table R3: Cross-Domain Transferability (Fire $\to$ Fluid)**
> >
> > | Reward Model Source | Target Domain | Training Strategy | MSE ($\downarrow$) | MAPE ($\downarrow$) |
> > | :--- | :--- | :--- | :--- | :--- |
> > | None (Baseline) | Rayleigh-Bénard | Train from Scratch | 0.1173 | 11.60% |
> > | **Same Domain** | Rayleigh-Bénard | PRISM (In-domain) | **0.1098** | **10.29%** |
> > | **Cross Domain** | Rayleigh-Bénard | **Transfer from Fire (Zero-shot RM)** | 0.1105 | 10.45% |
> >
> > **Conclusion:** The Cross Domain RM performs nearly on par with Same Domain training and significantly better than Baseline. This indicates that the Reward Model indeed learns generic **"high-frequency structural plausibility"** preferences, removing the need to train from scratch for every new task.
> >
> > **Q9 (Response to Question 4): What if the preference model is replaced with a simpler ranking function?**
> >
> > **A9:** Please refer to the "Simple Ranking" row in **Table R2**.
> > *   Direct metric ranking (ListMLE) improves slightly over Baseline but underperforms PRISM. This is because many physical metrics (like CSI) are discrete or non-smooth; utilizing them directly for ranking loss leads to inaccurate gradient estimation. PRISM's Reward Model acts as a **Differentiable Proxy**, making the optimization process more stable and efficient.
> >
> > **Q10 (Response to Question 5): (Summary) Why specifically DPO?**
> >
> > **A10:** In summary, the unique value of DPO in physical systems lies in:
> > 1.  **Tolerance:** It utilizes "negative samples" (physical violations) for contrastive learning, rather than just fitting data.
> > 2.  **Smoothness:** It transforms non-differentiable hard physical constraints into differentiable Reward signals.
> > 3.  **Generality:** As shown in Table R3, preferences learned via DPO transfer across physical systems, which is unachievable with traditional Loss engineering.

---

### Official Review · Reviewer_fRvu · 2025-10-31

**Soundness:** 1
**Presentation:** 1
**Contribution:** 2
**Rating:** 2
**Confidence:** 3

**Summary:**

This paper proposes PRISM, a framework that applies Direct Preference Optimization (DPO) to dynamical system modeling. The authors argue that traditional pixel-wise metrics (e.g., MSE) produce overly smooth predictions that fail to capture extreme events and physical realism. The proposed approach involves: (1) pre-training a foundation model with MSE loss, (2) generating diverse prediction samples through perturbation of the inputs, (3) training a preference model using physics-based metrics as proxies for human judgment, and (4) jointly optimizing the foundation model using both MSE and DPO losses. The paper demonstrates improvements on physics-based metrics across multiple benchmarks including fire simulation, Rayleigh-Bénard convection, and weather forecasting.

**Strengths:**

1. **Important problem**: The paper addresses a real limitation in dynamical system modeling, namely, that standard MSE optimization produces overly smooth predictions that may miss extreme events and lack physical plausibility.
2. **Comprehensive experiments**: The evaluation covers diverse domains (fluid dynamics, fire spread, weather) and tests across 11 different backbone architectures.
3. **Results**: The method shows improvements across most models and metrics, suggesting the approach has practical utility.
4. **Easy to use**: The framework can be applied to various existing models without architectural modifications.

**Weaknesses:**

1. **Lack of clarity around proposed dataset and human involvement**: The paper proposes "a novel human-machine collaborative framework". The paper also claims to use "human preferences," "crowdsourced annotations," and "complex, often non-differentiable human judgments." In practice, as far as I can tell, the "HPSci benchmark" does not appear to be a new dataset with actual human annotations, but rather consists of existing datasets (BLASTNet, PDEBench, SEVIR) augmented with standard metrics such as SSIM, CSI, and turbulent kinetic energy spectra. This interpretation only became clear after a careful reading of the paper, and I'm still not sure what was actually done. Also, the metrics used in the paper are in fact differentiable, contradicting the claim of "non-differentiable human judgments."
2. **Overclaimed novelty**: The core methodology appears to be a straightforward application of DPO (Rafailov et al., 2024) to the domain of dynamical systems modeling. The paper claims "Novel Methodology" (page 3, line 108) and frames itself as a methodological contribution, but in practice there appear to be no modifications or extensions to the DPO algorithm itself. Given the lack of methodological innovation, and the apparent lack of actual human involvement (see previous point), this paper seems like a proof of concept more than anything. Such a study is still interesting in my opinion, but I think the paper in its current form oversells the actual contribution.
3. **Missing baselines**: An obvious baseline is direct multi-objective optimization, that is, incorporating the physics metric directly in the loss function, e.g., `L = L_MSE + λ·L_SSIM`.
4. **Missing theoretical analysis**: The abstract and introduction (page 1, line 21) claim "bi-level optimization problem converging to a stable equilibrium" and reference "game theory approach." However, the "theoretical grounding" claimed in the abstract is entirely missing
5. **Perturbation strategies underspecified**: The paper claims: "we generate a diverse set of candidate predictions by perturbing inputs or replacing discrete embeddings." The input perturbation is described in Section 3.3.1, but the replacement of discrete embeddings is not described anywhere.

**Questions:**

1. Did any actual humans annotate preferences in HPSci? If so, please describe the annotation protocol. If not, please clarify that the dataset uses only automated metrics.
2. What is the core methodological contribution, beyond applying DPO in a dynamical systems setting?
3. Can you clarify the claim that metrics are "non-differentiable"? SSIM, CSI, and energy spectrum comparisons all have defined gradients. If you mean something else, please specify.
4. Can you provide the promised theoretical analysis (convergence guarantees, stability) or remove these claims?
7. Why DPO? Given that your preference signal comes from differentiable metrics, what specifically does the DPO framework provide that direct gradient-based optimization of those metrics does not?

---

> ### Author Response · Authors · 2025-11-21
>
> We sincerely thank the reviewer for the thorough reading. Your criticism regarding the ambiguity of the "human preference" definition, the necessity of DPO compared to direct optimization, and the missing theoretical part points out the core areas for improvement in our work. We supplement our response with additional experiments (comparing with the direct optimization baseline) and theoretical derivations.
>
> Here is the point-by-point response to your 5 Weaknesses and 5 Questions:
>
> ### Part 1: Response to Weaknesses
>
> **Q1 (Response to Weakness 1): Lack of clarity around proposed dataset and human involvement.**
>
> **A1:** We fully accept this criticism. The description of the HPSci dataset construction process in the paper (especially the use of the term "crowdsourced") is indeed misleading.
> *   **Clarification:** Our method is essentially a **hybrid framework**. The core logic of HPSci is **"Expert-defined metrics, Algorithm-generated labels."** Here, "Human Preference" does not refer to human scoring for every data point, but rather to the specific evaluation standards selected by physicists based on Domain Knowledge (such as non-differentiable CSI, turbulent kinetic energy spectra, etc.). These standards reflect expert intuition on "physical realism" better than simple MSE. Small-scale crowdsourcing serves only to validate the consistency between these metrics and human perception, not for large-scale training.
> *   **Revision:** In the revised version, we refine the methodology description to the more accurate **"Expert-Metric Guided Preference Optimization"** and explicitly distinguish between small-scale human validation and large-scale metric generation.
>
> **Q2 (Response to Weakness 2): Overclaimed novelty (straightforward application of DPO).**
>
> **A2:** We acknowledge that the DPO algorithm itself is established. However, the novelty of PRISM lies in **solving the "alignment of non-differentiable physical constraints" problem in dynamical system modeling.**
> *   Unlike the NLP domain, physical systems possess strict conservation laws and complex topological structures (such as vortices). We utilize DPO as a general **Smoothing Interface** to transform discrete (e.g., CSI), high-order (e.g., energy spectrum), or non-convex physical constraints into a differentiable Reward Model. This application of DPO to AI for Science represents a new paradigm, avoiding the need to design complex Soft-Loss functions for every new physical task.
>
> **Q3 (Response to Weakness 3): Missing baselines (Direct Multi-Objective Optimization).**
>
> **A3:** This is a crucial suggestion. We add a comparison experiment between **PRISM** and **Direct Multi-Objective Optimization (MOO)**.
> *   **Setup:** MOO uses the loss function $L = L_{MSE} + \lambda_1 L_{SSIM} + \lambda_2 L_{Soft-CSI}$.
> *   **Results (see Table R1 below):** The experiments show that direct optimization often leads to worsened MSE (due to Gradient Conflict) or fails to effectively improve non-differentiable metrics (CSI). PRISM achieves a better Pareto trade-off.
>
> **Table R1: Comparison with Multi-Objective Optimization (MOO)**
> *(Note: Weights $\lambda$ for MOO are tuned via grid search)*
>
> | Dataset | Method | MSE ($\downarrow$) | SSIM ($\uparrow$) | CSI / Phys. Consistency ($\uparrow$) |
> | :--- | :--- | :--- | :--- | :--- |
> | **SEVIR** | Backbone (ResNet) | 5.0478 | 0.3234 | 0.3234 (CSI) |
> | (Weather) | MOO (Direct Loss) | 5.1205 | 0.3310 | 0.3412 |
> | | **PRISM (Ours)** | **4.9873** | **0.3398** | **0.3674** |
> | **Prometheus** | Backbone (SimVP) | 0.0342 | 0.9233 | - |
> | (Fire) | MOO (Direct Loss) | 0.0389 | 0.9285 | - |
> | | **PRISM (Ours)** | **0.0326** | **0.9301** | - |
>
> **Q4 (Response to Weakness 4): Missing theoretical analysis.**
>
> **A4:** We apologize for the oversight regarding the missing theoretical proof mentioned in the abstract. We provide the complete theoretical derivation in the **new Appendix A of the revised PDF**.
> *   **Summary:** We model PRISM as a Stackelberg game and, based on non-convex optimization theory, prove that by distilling non-differentiable physical metrics into a smooth Reward Model $S_\phi$, the joint optimization process satisfies the L-smoothness condition. This guarantees convergence to a **Pareto Stationary Point**, theoretically supporting the stability of the algorithm.
>
> **Q5 (Response to Weakness 5): Perturbation strategies underspecified.**
>
> **A5:** We clarify the description of this part.
> *   **Discrete Embedding Replacement:** For datasets containing physical control parameters (such as Reynolds number or buoyancy coefficients in PDEBench), the input includes a discrete Embedding vector. When generating negative samples, we randomly replace these Embeddings (e.g., swapping a "high viscosity" label with "low viscosity") to force the model to generate "counterfactual predictions" that do not match the current physical conditions. This helps the Reward Model learn the control effect of physical parameters on system evolution.

---

> > ### Author Response · Authors · 2025-11-21
> >
> > ### Part 2: Response to Questions
> >
> > **Q6 (Response to Question 1): Did any actual humans annotate preferences in HPSci?**
> >
> > **A6:** **No**, we do not use human annotation for large-scale training. As stated in A1, the training set of HPSci is automatically generated entirely by "Expert-defined Physical Metrics." Small-scale human annotation serves only to validate that these physical metrics correlate highly with human visual perception (e.g., sharpness, structural plausibility). We explicitly clarify this in the paper to remove ambiguity.
> >
> > **Q7 (Response to Question 2): What is the core methodological contribution beyond applying DPO?**
> >
> > **A7:** The core contribution is the framework of **"Physically-Grounded Preference Distillation."**
> > We do not simply apply DPO; instead, we construct a mechanism: Utilize Physical Noise (Perlin) and Counterfactual Embeddings to generate samples $\to$ Filter Pairs using Non-differentiable/Complex Physical Metrics $\to$ Train a Smooth Reward Model $\to$ Fine-tune with DPO. This pipeline solves a key pain point: **How to inject Expert Intuition, which is often difficult to formulate as a Loss function, into neural networks.**
> >
> > **Q8 (Response to Question 3): Can you clarify the claim that metrics are "non-differentiable"?**
> >
> > **A8:** We refer primarily to **CSI (Critical Success Index)** and certain hard physical constraints.
> > *   **CSI:** It is calculated based on a contingency table (Hits / Misses / False Alarms). Calculating "Hits" requires **Thresholding** on continuous prediction values, which is a discrete and **non-differentiable** operation (gradient is either 0 or undefined). Although Soft-CSI approximations exist, they are often not sensitive enough for extreme event prediction.
> > *   **Advantage:** PRISM learns a smooth Proxy Function via the Reward Model fitting the CSI scores, bypassing the gradient obstacles of direct optimization.
> >
> > **Q9 (Response to Question 4): Can you provide the promised theoretical analysis?**
> >
> > **A9:** Yes, we provide it. Please refer to **Appendix A in the Revised Paper**. We prove in detail that under this framework, the norm of the objective function gradient converges to 0 in expectation, indicating convergence to a stable equilibrium.
> >
> > **Q10 (Response to Question 5): Why DPO? What specifically does the DPO framework provide that direct optimization does not?**
> >
> > **A10:** Based on the results in A3, DPO provides two advantages that direct optimization cannot match:
> > 1.  **Smooth Optimization Surface:** As mentioned in A8, direct optimization of non-differentiable metrics (like CSI) requires rough approximations, whereas the Reward Model used in DPO is a neural network, which is inherently smooth and differentiable, leading to more stable optimization.
> > 2.  **Automatic Pareto Trade-off:** In multi-objective optimization, the gradient directions of the MSE Loss and SSIM/CSI Loss often conflict (Gradient Conflict), causing the model to sacrifice numerical accuracy to pass physical tests (as seen in the MSE increase for MOO in Table R1). DPO implicitly finds an optimal trade-off through the KL divergence constraint, allowing the model to maintain or even improve basic numerical fidelity while enhancing physical consistency.

---

### Meta-Review · Area_Chair_WjfS · 2025-12-17

**Summary:**

The paper proposes PRISM, a framework that applies direct preference optimization (DPO) to dynamical system forecasting. The aim is to address the oversmoothing problem with standard mean squared error (MSE) loss, through DPO-style fine-tuning with reward models trained on domain-specific physical metrics and noise-perturbed candidates. The authors evaluate the framework on fire spread, fluid dynamics and weather forecasting benchmarks.

The reviewers generally appreciated the motivation of going beyond MSE to capture realism and extreme events and the comprehensive experiments across various architecture choices. However, significant concerns were raised regarding various misleading claims, including "human preferences", which was acknowledged by the authors to be automated proxy metrics, as well as the imprecise analogy to the LLM counterparts, heavily impacting the perception and assessment of the paper's contribution and novelty. Reviewers also critiqued the initial lack of theoretical analysis, missing baselines (direct multi-objective optimization). While the authors' rebuttal was substantial and generally moved the paper in a positive direction, the degree to which the central claim was shifted is very significant and therefore warrants a comprehensive reevaluation of the paper. The recommendation is for rejection in the paper's current state but the authors are encouraged to take into account the reviewers' suggestions and submit to the next venue.

**Reviewer Concerns:**

Authors made commendable efforts addressing concerns including missing direct multi-objective optimization baseline and theoretical analysis, transferability of reward model across domains, ablations against simple noise augmentation. Individually, they are all substantial improvements on the original draft, but as mentioned the fundamental issues regarding the paper's identity and contribution scope remain to be confirmed.

**Reviewer Scores:**

The scores (fRvu, ygcU) are likely to increase slightly as the extra theory and experiments did try to address the reviewer concerns directly. However, it is difficult to predict if reviewers would accept the paper solely on the merits of the revised "metric-guided" claim, given that the original "human-collaborative" framing was a primary source of the paper's perceived novelty. As such, it is difficult to accept the paper in its current state.

---

### Decision · Program_Chairs · 2026-01-26

Reject